# Curiosity-Driven Learning of Joint Locomotion and Manipulation Tasks

**Clemens Schwarke,**\* **Victor Klemm,**\* **Matthijs van der Boon,**
**Marko Bjelonic,** **and Marco Hutter**
Robotic Systems Lab, ETH Zurich, Switzerland
{cschwarke, vklemm, mvander, markob, mahutter}@ethz.ch

**Abstract:** Learning complex locomotion and manipulation tasks presents significant challenges, often requiring extensive engineering of, e.g., reward functions or curricula to provide meaningful feedback to the Reinforcement Learning (RL) algorithm. This paper proposes an intrinsically motivated RL approach to reduce task-specific engineering. The desired task is encoded in a single sparse reward, i.e., a reward of "+1" is given if the task is achieved. Intrinsic motivation enables learning by guiding exploration toward the sparse reward signal. Specifically, we adapt the idea of Random Network Distillation (RND) to the robotics domain to learn holistic motion control policies involving simultaneous locomotion and manipulation. We investigate opening doors as an exemplary task for robotic applications. A second task involving package manipulation from a table to a bin highlights the generalization capabilities of the presented approach. Finally, the resulting RL policies are executed in real-world experiments on a wheeled-legged robot in biped mode. We experienced no failure in our experiments, which consisted of opening push doors (over 15 times in a row) and manipulating packages (over 5 times in a row).

**Keywords:** Curiosity, Reinforcement Learning, Wheeled-Legged Robots

## 1 Introduction

Recent advancements in Reinforcement Learning (RL) have propelled legged and wheeled-legged robots beyond the confines of the lab, empowering them to fulfill a broad range of practical applications [1, 2, 3, 4, 5, 6]. Nevertheless, tasks in environments designed for human interaction remain challenging as they frequently require coordination between locomotion and manipulation. Research often aims to reduce task complexity by manually dividing the problem into multiple stages [7]. However, this approach lacks the exploration of holistic solutions that consider the interplay between different stages. Solving these problems end-to-end with minimal task-specific engineering, i.e., without including a variety of dense reward terms that need to be tuned extensively, remains an open challenge.

Figure 1: A wheeled-legged robot picking up a package and opening a door while balancing on two wheels. Motions can be seen at https://youtu.be/Qob2k_ldLuw.

---

\*Shared first authorship.

7th Conference on Robot Learning (CoRL 2023), Atlanta, USA.

This work proposes an RL approach that solves the task as a whole with little need for task-specific engineering. We adopt a sparse reward setting, rewarding "+1" when the desired task is achieved. Although simple in its formulation, the discovery of sparse rewards is in general challenging and requires strategies for the exploration of the environment. Because the reward only exists in a small fraction of the state space, random exploration induced by, e.g., epsilon-greedy policies is not sufficient to pick up the reward signal in reasonable time [8]. In our work, intrinsic motivation is employed to guide the agent. For the first time, the concept of Random Network Distillation (RND) [9], which models intrinsic motivation, has been successfully applied and validated in a real-world robotic task, as shown in Fig. 1. Our main contributions can be summarized as follows:

1. A curiosity-driven sparse reward RL approach for learning end-to-end locomotion and manipulation tasks without laborious, task-specific engineering (Section 3)

2. The notion of curiosity states as guiding mechanisms, allowing to focus curiosity on non-directly observable states, while maintaining simple deployment (Section 2.2)

3. An analysis of emerging learning behaviors (Section 4.1)

4. Successful real-world validation of RL policies on a wheeled-legged robot in biped mode [10], by repeatedly opening a door and manipulating a package (Section 4.2)

## 1.1 Mechanisms to Guide Exploration

We group existing approaches that employ guided exploration to improve learning performance over random exploration into three main categories. **Expert Demonstrations** can be an effective tool to teach a desired skill [11, 12, 10, 13], but require predominantly hand-crafted demonstrations. **Curriculum Learning** [14, 15, 16, 17, 18] involves gradually increasing task difficulty during training, but the generation and efficient scheduling of intermediate tasks are still considered unsolved and subject to ongoing research [17, 19, 20]. **Intrinsic Motivation**, more specifically curiosity, denotes the ability to learn without external rewards for the pure sake of knowledge gain. Integrating this mechanism into our learning algorithms holds great promise, particularly due to its task-agnostic nature, which distinguishes it from expert demonstrations or curriculum learning.

Past research has explored three main approaches to modeling intrinsic motivation. In an initial endeavor to incorporate curiosity into an RL algorithm, [21] introduced a reward based on the Euclidean distance between the prediction of a model learning the environmental forward dynamics and the observed transition, effectively **rewarding surprise**. The concept of measuring novelty using prediction error from a dynamics model raises a significant concern: it tends to favor stochastic transitions [22], as they are difficult or even impossible to predict accurately. To address this challenge, [23] adopts a different approach by predicting a state embedding instead of the complete world state. Another line of work focuses on estimating and **rewarding learning progress** within specific regions of the state space [24, 25, 26]. However, measuring learning progress in high dimensional continuous state spaces remains computationally infeasible [23]. Count-based exploration methods work by **rewarding state novelty** directly by keeping track of the number of visits for each state and prioritizing less visited states [27]. To tackle exploration in continuous state spaces, [28] proposes pseudo-counts, a generalization of count-based exploration methods. Burda et al. [9] propose RND, a method based on predicting information about the current state to measure state novelty. They train a prediction model alongside the RL agent in a supervised fashion, improving its accuracy for visited states during the learning process. The prediction error can then be utilized as an intrinsic reward signal, where familiar states yield more accurate predictions compared to less visited or unvisited states. Given our focus on loco-manipulation tasks characterized by continuous state spaces and potentially complex task dynamics, we consider the third option as the most suitable.

## 1.2 Loco-Manipulation

In recent years, approaches to solving complex loco-manipulation tasks are dominated by RL, e.g., [6, 29, 30], with some works fully or partly relying on model predictive control (MPC) [31, 32].

A common design choice to break down the complexity of loco-manipulation is to divide the problem into a locomotion and a manipulation task and to control them individually [32, 33]. This introduces new challenges in engineering the communication between both controllers and leads to sub-optimal behaviors as synergies between different body parts can not be fully exploited [34]. Other works focus on manipulating objects through locomotion, e.g., pushing an object through walking forward [31, 35, 36]. More dynamic manipulation tasks are investigated by a different body of work that studies soccer skills [30, 37, 38, 6]. Many of these works combine multiple low-level skills into a more capable controller, mostly through a hierarchical control framework and a skill library [30, 38, 35, 37]. This could be a future research direction to combine the controllers proposed in this work. However, little work exists on how to leverage intrinsic motivation to learn complex loco-manipulation tasks in a lean task-independent framework.

In this study, the primary task under examination is opening doors. A quadrupedal robot by Boston Dynamics equipped with an arm has demonstrated impressive performance in the task of opening doors [39]. However, limited information is available regarding the specific approach employed, only that it is model-based. Other works divide the door opening task into sub-tasks that can be solved in sequence [40, 7]. This requires task-specific modeling and tailoring of the control scheme, which can be identified as a common shortcoming in many of the mentioned works. In the following, we aim for a more holistic approach.

## 2 Curiosity Formulation

Structured exploration is a crucial factor for successful learning in sparse reward settings. RND [9] offers an intuitive and computationally efficient approach to intrinsic motivation, while the curiosity state proposed in this work focuses exploration.

### 2.1 Random Network Distillation

The RND module consists of two function approximators. A randomly initialized target network $\boldsymbol{f}$ encodes states $\boldsymbol{s} \in \mathcal{S}$ into an unknown embedding $\boldsymbol{f}(\boldsymbol{s})$. The target stays fixed during the whole training process. A predictor network $\hat{\boldsymbol{f}}$ estimates the target's embedding, given the same input $\boldsymbol{s}$ as the target. The predictor network is trained alongside the RL agent on the visited states in a supervised fashion with a Mean Squared Error (MSE) loss. The prediction error, i.e., the difference between the outputs of both networks serves as the intrinsic reward signal defined by

$$r_{\text{intrinsic}} = \left\| \boldsymbol{f}(\boldsymbol{s}) - \hat{\boldsymbol{f}}(\boldsymbol{s}) \right\|_2 . \tag{1}$$

Intuitively, familiar state regions yield small prediction errors as the predictor is already trained on similar states. Not yet visited state regions lead to large errors and therefore large intrinsic rewards. As the agent visits an unfamiliar region repeatedly, the prediction error decreases.

### 2.2 Curiosity State

While [9] applies the RND module directly to the agent's observations $\boldsymbol{o}$ instead of the state $\boldsymbol{s}$, we propose a more flexible formulation where a *curiosity state* $\boldsymbol{s}_c = \phi(\boldsymbol{s})$ is passed to the RND module to stay independent of the observations. The mapping $\phi$ can be freely chosen, as long as the state $\boldsymbol{s}$ can be implicitly inferred from the environment's feedback. This way, curiosity can be focused on the desired quantities, even though they might not be directly observable during deployment of the motion policy. The thereby introduced flexibility allows to leverage the curiosity module in simulation while keeping deployment simple, thus showing a practical adaption of the RND formulation to the robotics domain. While this formulation allows for arbitrary mappings $\phi$, it suffices to consider $\phi$ to select a subset of the full state without further modification. A sketch of the implemented RND module is shown in Fig. 2.

# 3 Task-Specific RL Formulation

We illustrate an effective sparse reward formulation on the exemplary task of door opening, detailing the chosen rewards, observations, and the introduced curiosity state. Subsequently, we adapt the formulation to the task of package manipulation to demonstrate the straightforward generalization of the proposed approach to different tasks.

## 3.1 Rewards

The chosen reward function consists of the three reward terms $r = r_{\text{intrinsic}} + r_{\text{task}} + r_{\text{shaping}}$. The first term $r_{\text{intrinsic}}$ is defined by equation 1 and motivates the agent to explore the relevant part of the state space. In this work, we use Multi Layer Perceptrons (MLPs) with 1 and 2 hidden layers with 5 neurons and a one-dimensional output for the target and predictor network, respectively. The second term $r_{\text{task}}$ is the only task-specific reward. For the task of door opening, it is defined intuitively by repeatedly giving a reward of +1 while the task is achieved, i.e., when the door hinge angle $q_{\text{hinge}}$ is in a desired interval, and no reward otherwise:

$$r_{\text{task}} = \begin{cases} 1, & \text{if } 1.5 < q_{\text{hinge}} < 2.1 \\ 0, & \text{otherwise} \end{cases} . \quad (2)$$

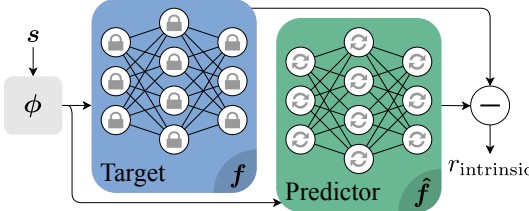

Figure 2: Random Network Distillation (RND) module. A mapping $\phi$ encodes the full system state $s$ into the curiosity state $s_c$. The target network with fixed weights (blue) embeds the curiosity state $s_c$ into a latent representation. The predictor network (green) is updated during training and attempts to match the target's output. The difference between outputs serves as the intrinsic reward signal.

The inclusion of the third term $r_{\text{shaping}}$ serves as an incentive for the robot to maintain a standing posture and adopt more refined, less forceful strategies. This emphasis on smoother and less aggressive policies is crucial for achieving effective sim-to-real deployment within the realm of RL control [15, 41]. For a comprehensive breakdown of the rewards and their respective weights, please refer to Appendix A.1.

## 3.2 Observations

In terms of observations, our approach focuses on minimizing task dependency and avoids the use of complex exteroceptive information. The only observation that pertains specifically to the door is the relative position $_C r_{CH}$ of the door handle origin in the robot's camera frame. We freeze the initial door handle position and provide it as an additional input $_C r_{CH_{\text{init}}}$ to the policy to determine the degree to which the door is open.

We provide a list of standard observations that are not task-specific and have been covered in previous works [10, 4] in Appendix A.1. We note that the observations are subject to empirical normalization for proper scaling of input variables.

## 3.3 Curiosity Implementation

An advantage of the proposed curiosity state $s_c$ is its independence of the robot's observations. Since the RND module is only needed for training, it is possible to include quantities that are easy to attain in simulation but hard to estimate in reality and thus unfit as observations. To focus the agent's curiosity on the door, we include door hinge and handle angles $q_{\text{door}}$ as well as their angular velocities $\dot{q}_{\text{door}}$ in the curiosity state. Adding the distance between the robot and the door handle $d_{CH}$ induces faster interaction with the door. To avoid an intrinsic reward signal for moving too far away from the door, the distance is clipped according to $d_{CH} = \min(\|r_{CH}\|_2, 2)$, and the curiosity state is defined as $s_c = \begin{bmatrix} q_{\text{door}}^\top & \dot{q}_{\text{door}}^\top & d_{CH} \end{bmatrix}^\top$.

### 3.4 Generalization to Package Manipulation

We show the task independence and generalization capability of the proposed approach by subjecting it to a second task requiring different locomotion and manipulation skills. We choose the exemplary task of package manipulation, i.e. grabbing, moving, and dropping a package, as it involves a freely moving object, as opposed to the fixed-base articulated door. To encode the task, we define a similar sparse task reward $r_{\text{task}}$, given by

$$r_{\text{task}} = \begin{cases} 1, & \text{if } _{\mathcal{I}}\boldsymbol{r}_{\text{package}} \in \mathcal{S}_{\text{bin}} \\ 0, & \text{otherwise} \end{cases}, \tag{3}$$

where $_{\mathcal{I}}\boldsymbol{r}_{\text{package}}$ is the position of the package in the inertial frame $\mathcal{I}$ and $\mathcal{S}_{\text{bin}}$ includes the space in and above the bin. We include the space above the bin to reduce reward sparsity as the agent still learns to drop the package. It is however not necessary to include this space for successful learning. Instead of observing the door handle position, the agent now observes the relative package, table, and bin positions $_{\mathcal{C}}\boldsymbol{r}_{CP}$, $_{\mathcal{C}}\boldsymbol{r}_{CT}$, and $_{\mathcal{C}}\boldsymbol{r}_{CB}$, respectively. As before, relative positions are observed in the camera frame. The last necessary change concerns the curiosity state. We define it as $\boldsymbol{s}_c = \begin{bmatrix} _{\mathcal{I}}\dot{\boldsymbol{r}}_{\text{package}}^{\top} & d_{PB} & d_{CP} \end{bmatrix}^{\top}$, where $_{\mathcal{I}}\dot{\boldsymbol{r}}_{\text{package}}$ is the linear package velocity in the inertial frame. The distances between package and bin $d_{PB}$ and camera and package $d_{CP}$ are again clipped.

## 4 Experimental Validation and Discussion

To validate our approach, we conduct experiments in simulation and the real world. For implementation details please refer to the Appendix. All experiments are conducted with the wheeled-legged quadrupedal robot in Fig.1, a dynamic robot that can perform hybrid motions between walking and driving. Recently, [10] discovered a bipedal locomotion mode through RL, further increasing the robot's versatility. In bipedal mode, the front legs can serve as arms to manipulate objects.

### 4.1 Simulation Results

Qualitative examples of the door opening and package-grabbing motions can be seen in the supplementary video.[2] In Fig. 4, we present quantitative results for all three tasks with different reward settings. In general, we noticed that there is a divide between runs in which the agent learns to

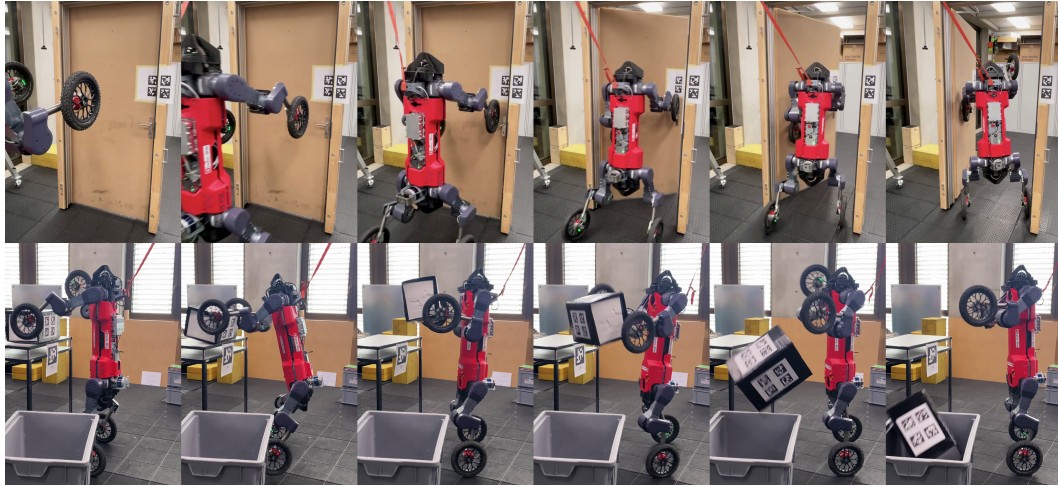

Figure 3: Snapshot sequences (left to right) of the push door and package manipulation experiments. Opening the door takes 2.5 seconds, grasping and dropping the package 1.5 seconds.

---

[2]To enable pull door experiments, we equip the robot with basic hooks attached to the front wheels.

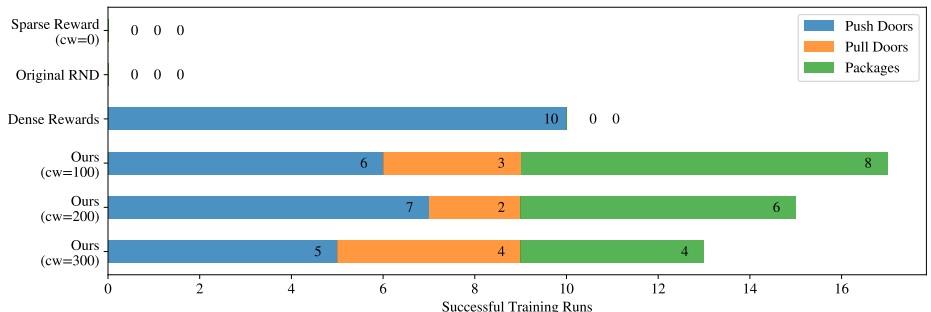

Figure 4: Number of training runs (out of 10 for each task) in which the agent successfully learned to accomplish the given task for different reward formulations and different intrinsic reward weights (denoted as cw). Every experiment was conducted for random seeds 1-10.

complete a given task and others where the sparse reward is solely triggered through randomness. Only in rare cases, the sparse reward is discovered but remains small due to shaping rewards that hinder skill discovery. We consider training runs with a final success rate greater than $25\%$ successful to exclude the aforementioned cases. Successful runs show success rates significantly above this threshold, as can be seen in Appendix A.3, where we also provide a more thorough analysis for the case of (out-of-distribution) environment disturbances of various magnitudes. The best-performing policies achieve success rates of $99\%$, $92\%$, and $99\%$, for the push door, pull door, and package manipulation tasks, respectively. In the following, we discuss the relevant findings of learning with curiosity.

**Training Evaluation:** The need for guided exploration becomes clear when training with extrinsic reward signals only, as reported in Fig. 4. Indeed, the investigated skills are not learned. Instead, the learning process ends in a local optimum and the agent learns to stand without moving. The added intrinsic reward signal guides the agent toward states that involve manipulating the door or package. An exemplary learning process is shown in Fig. 5 for a push door. The learning process for package manipulation evolves similarly. First, the robot discovers the package and moves it on the table to increase the intrinsic reward signal. After learning to grab the package with both wheels, the robot starts to wiggle the package and moves it closer to the bin. Once the sparse task reward is found, the intrinsic reward decreases as the agent optimizes its behavior to achieve the desired task.

**RND Evaluation:** Experiments show that the network architecture of the RND module can be kept minimal. While a predictor of the same size as the target should be able to fully approximate the target, choosing a larger predictor leads to more consistent intrinsic reward curves over different training runs. The weight of the intrinsic reward does not require extensive tuning since learning is successful for a large interval, as can be seen in Fig. 4. Normalizing the reward empirically as in [9] does not improve training and prevents the reward from converging to values close to zero. Again, convergence to small values is necessary to model the loss of interest as the agent gets more familiar with the environment. Normalization scales the reward with its standard deviation and thus leads to larger rewards at the end of training. In contrast, normalization of the curiosity state is crucial for a proper decay of the intrinsic reward over the course of training. The curiosity state includes quantities that can reach high magnitudes like velocities and is therefore subjected to empirical normalization, i.e., it is normalized with a running estimate of its mean and variance. While normalization improves the convergence properties of the intrinsic reward, it violates the theoretical foundation of state visitation-based curiosity. During training, the same state keeps changing its normalized representation due to varying normalization parameters. Different states could have almost identical representations for different points in time and thus also almost identical target embeddings. Even though this distorts the measure for state familiarity, curiosity state normalization does not seem to have a negative influence on exploration and is highly recommended to improve convergence to a small reward signal.

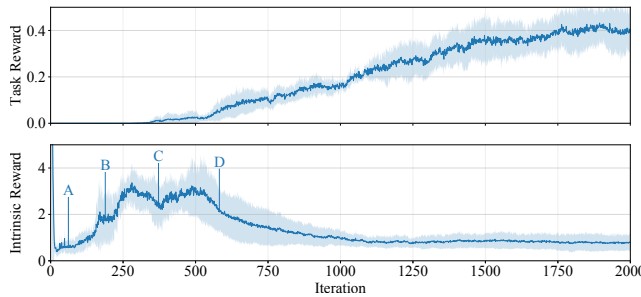
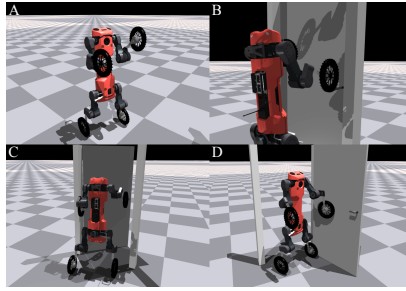

Figure 5: Task specific reward $r_{task}$ and intrinsic reward $r_{intrinsic}$ over the course of learning to open a push door. The mean and standard deviation include 4 successful out of 5 training runs for a curiosity weight of 100 and random seeds 1-5. After learning how to stand (A), the robot starts to play with the door handle and the intrinsic reward increases (B). Next, the robot opens the door slightly while still manipulating the door handle (C). The agent then discovers the sparse door opening reward and starts to optimize toward it (D).

**Curiosity State Evaluation:** To assess the relevance of the curiosity state notion, we probe the original RND formulation of [9], that applies the curiosity module to the entirety of the observation space. We note that skill discovery is not feasible for the investigated tasks due to two reasons. First, the large state space might cause an unreasonable amount of exploration to properly converge in reasonable time. Second, parts of the state space that would be crucial to explore might not be directly observable, e.g., the door handle angle. Using a curiosity state instead of the observation space allows the use of privileged information which might not be directly observable in the real world, but is available in simulation.

**Emerging Behaviors:** A key finding of this paper is the sensitivity of the learning process to small changes in the training environment. Changing the seed used to initialize networks or set randomized quantities is enough to alter the resulting policy significantly. RL approaches usually rely on dense task rewards that heuristically steer the agent toward a feasible trajectory. This might bias the agent toward suboptimal behavior preventing discovery of the sparse reward. We evaluate a naive implementation of a dense reward setup of comparable complexity to our method. For simple tasks like opening push doors, the dense reward setup is able to discover the skill, but for the other, more complex tasks it is not. In contrast to a dense task reward setting, a sparse task reward setting does not bias the agent toward any trajectory. Randomness in the exploration can thus lead to the discovery of different minima and trajectories, especially in contact-rich scenarios. As a result, different behaviors emerge as shown in Fig. 6 where the robot is holding the door open in a variety of poses. For the task of package manipulation, differences are more subtle. During task execution, policies mainly differ in the leg configuration and the stepping pattern of the robot.

## 4.2 Real-World Results

We demonstrate the effectiveness of the proposed approach by opening a push door over 15 times in a row without a single failure in a lab environment. Learned policies are able to let the robot stand and navigate toward the door. The robot reaches for the door handle with its right wheel and attempts to press it while pushing against the door. As soon as the door is unlocked, the robot swings the door open and holds it open while standing still, as shown in the top row of Fig. 3. Sim-to-real transfer benefits from modeling the robot's Field of View (FOV) in training, explained in Appendix A.2. Since the real robot lacks the added hooks for pulling doors, we leave pull-door experiments to future work. For the task of package manipulation (shown in the bottom row of Fig. 3), the robot robustly grabs the package and drops it into the bin over 5 times in a row. In one instance out of all tests, the robot lost its grip on the package. However, the robot quickly regrabbed the package and successfully delivered it. Policies exhibit highly dynamic behavior. If a more gentle motion is desired, a task reward that does not favor quick task completion would be more appropriate.

### 4.3 Limitations

Depending on the desired task, the intrinsic motivation approach might come with some trade-offs. On one hand, the sparsity of rewards encourages the exploration of diverse behaviors (see Fig. 6), which can lead to new solutions, often neglected by more dense approaches. On the other hand, it complicates the process of enforcing a specific behavior. Although dense shaping rewards can be added to enforce the desired behavior, they limit exploration and reduce training robustness. This can be seen in Fig. 4, where the investigated skills are not discovered in every training run.

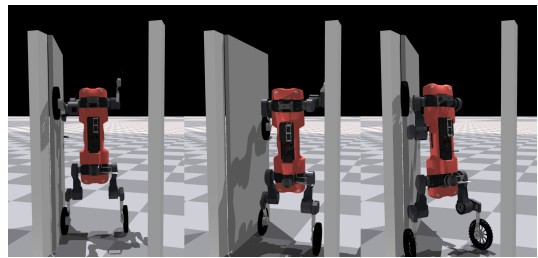

Figure 6: A wheeled-legged robot holding the door open in a variety of poses.

Secondly, since the intrinsic reward signal is not vanishing completely, the agent stays curious about its environment even after finding the task reward. In our case, this did not cause unwanted behaviors. If it does for other applications of RND, the weight of the reward could be scheduled.

Lastly, as the predictor network is trained in a supervised fashion, overfitting to specific regions of the state space could occur. Although not observed in this work, this might be exploited by the RL agent in a back-and-forth fashion. Subsequent switching to different state space regions would gain intrinsic reward repeatedly. Common methods to avoid overfitting such as regularization or dropouts could be employed in that case.

## 5 Conclusions and Future Work

We show that intrinsic motivation for exploration proves successful in simulation, yielding motion policies for complex tasks that involve locomotion and manipulation. Our RL method proves to generalize over multiple tasks requiring different sets of skills. We note that different behaviors emerge for small changes in the training environment. This phenomenon is explained by the absence of dense task rewards that bias the agent toward specific trajectories and is inherent to the sparse reward setting.

The introduced notion of a curiosity state guides exploration toward the reward in an efficient manner and allows learning of various tasks with basic task-dependent observations. We note that curiosity state normalization is crucial for proper reward convergence during the course of training, even though it distorts the measure of state familiarity.

To validate the proposed method, trained motion policies are executed on a wheeled-legged robot in biped mode. Experiments show that the robot is able to successfully and robustly open a push door in a lab environment, over 15 times in a row without failure, as well as manipulate a package through a grabbing, moving, and dropping motion over 5 times in a row without failure. We conclude that the investigated approach proves valuable for the robotics control domain as it enables the learning of highly complex skills with a minimal amount of task-specific engineering.

Future research could involve further investigation into the chosen curiosity formulation, the notion of penalty-based surprise [42] could allow for gentle policies without the need for task-specific shaping rewards. Other potential continuations include investigating controllers that achieve multiple tasks by combining the control policies proposed in this work.

### Acknowledgments

This project has received support from the European Union's Horizon Europe and H202 Framework Programme under grants agreement No. 101070596 and No. 852044 as well as from the Swiss National Science Foundation (SNSF) as part of project No. 166232.

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

# Appendix

In the following, we provide implementation details of the simulation and real-world experiments, as well as further quantitative evaluations of the investigated approach.

## A.1  Simulation Setup

We train with NVIDIA's Isaac Gym [43] and employ Proximal Policy Optimization (PPO) [44]. A detailed description of the used training pipeline can be found in [15]. A full training run comprises 2000 policy updates to ensure reward convergence for all investigated tasks. It takes one hour to train a policy on a single NVIDIA RTX 2080 Ti graphics card. Subsequently, we give a detailed description of the training environment.

**Reward Formulation:** The definitions and weights of the reward terms used for the door and the package task are detailed in Table 1. We decided to add two task-related shaping rewards for the task of package manipulation to improve the behavior for real-world tests. Namely, the agent receives penalties for generating high package velocities and exerting large contact forces onto the table. Notice that this choice does not violate the idea of the proposed approach. Firstly, the added penalties are unrelated to the main task, which is still defined by a single sparse reward. Secondly, our approach first generates unbiased behaviors and can then be augmented for more pleasing results. In contrast, other formulations bias the agent as a byproduct of defining the desired task in a dense fashion. Penalizing table contacts and the package velocity, which is part of the chosen curiosity state, clearly increases the difficulty of discovering the desired skill. To compensate for this, we employ a simple reward scaling scheme. The first 1000 training iterations serve as a discovery phase, as most runs discover the sparse reward in that time. Shaping and standing rewards are active but scaled by a factor of 0.1. The second half of training acts as a shaping phase where the scaling factor is gradually increased to 1 over the course of 500 iterations.

**Observations:** The corresponding observation definitions can be found in Table 2. All observations are subject to noise to account for uncertainties and sensor noise in reality. For more detail in that regard, please refer to [15].

**Randomization:** To improve generalization to different environments, as well as robustness against mismatches between simulation and reality, masses and friction coefficients are randomized as detailed in Table 3. Additionally, the robot spawns in a randomized pose, i.e., initial position, orientation, and joint configuration vary. All randomized properties are sampled from a uniform distribution in the interval of $[\mu - \frac{\epsilon}{2}, \mu + \frac{\epsilon}{2}]$ for every training environment.

**Termination Conditions:** Episodes terminate after 8 seconds, resetting the environments to their initial state. An episode terminates early if either the robot is in collision, or if the robot's center is too low, i.e., if the robot does not manage to stand and falls. The second condition accelerates training but is not necessary for successful learning. We also terminate an episode if the package is not in contact with either the table or the front wheels to prevent the agent from directly throwing the package. This termination condition is disabled in close proximity to the bin to allow the dropping of the package into the bin.

**Door Model:** The considered doors feature standard lever door handles that need to be pressed to a certain degree to unlock the door. In simulation, the handle needs to be pressed once to keep the door unlocked for the rest of the episode. Dynamics of the hinge and handle are modeled as spring-damper systems with a constant torque offset $\tau_{\mathrm{const}}$. This is achieved by applying the torque

$$\boldsymbol{\tau}_{\mathrm{door}} = \boldsymbol{\tau}_{\mathrm{const}} + \mathrm{diag}(\boldsymbol{k}) \cdot \boldsymbol{q}_{\mathrm{door}} + \mathrm{diag}(\boldsymbol{d}) \cdot \dot{\boldsymbol{q}}_{\mathrm{door}}, \qquad (4)$$

to the door joints. Constants $\boldsymbol{\tau}_{\mathrm{const}}$, $\boldsymbol{k}$, and $\boldsymbol{d}$ are randomized by sampling from a uniform distribution. Measurements on the lab door provide reference values for realistic door dynamics. Further details are provided in Table 3.

**Field of View Simulation:** To mimic the perception system of the real robot we simulate the FOV for egocentric vision, as introduced in simulation experiments in [29], resulting in behaviors that

Table 1: Rewards

| Name | Formula | Weight |
|---|---|---|
| **Intrinsic Reward** | | |
| RND prediction error | $\left\|f(\boldsymbol{s}_c) - \hat{f}(\boldsymbol{s}_c)\right\|_2$ | 100 |
| **Task Rewards** | | |
| Door opened | $\begin{cases} 1, & \text{if } 1.5 < q_{\text{hinge}} < 2.1 \\ 0, & \text{otherwise} \end{cases}$ | 1.0 |
| Package delivered | $\begin{cases} 1, & \text{if } {}_{\mathcal{I}}\boldsymbol{r}_{\text{package}} \in \mathcal{S}_{\text{bin}} \\ 0, & \text{otherwise} \end{cases}$ | 1.0 |
| **Standing Rewards** | | |
| Height | ${}_{\mathcal{I}}z_{\text{base}}$ | 0.5 |
| Upright base | $\frac{\pi/2 - \arccos({}_{\mathcal{I}}\boldsymbol{e}_x^{\mathcal{B}} \cdot {}_{\mathcal{I}}\boldsymbol{e}_z^{\mathcal{I}})}{\pi/2}$ | 0.5 |
| Straight shoulder joints | $-\|\boldsymbol{q}_{\text{shoulders}}\|^2$ | 0.5 |
| Straight knee joints | $\exp(-\|\boldsymbol{q}_{\text{knees}}\|^2)$ | 0.25 |
| **Shaping Rewards** | | |
| Joint torque | $-\|\boldsymbol{\tau}\|^2$ | $1.5 \cdot 10^{-5}$ |
| Joint acceleration | $-\|\ddot{\boldsymbol{q}}\|^2$ | $2.5 \cdot 10^{-7}$ |
| Joint velocity | $-\|\dot{\boldsymbol{q}}\|^2$ | $2.5 \cdot 10^{-4}$ |
| Action difference | $-\|\boldsymbol{a} - \boldsymbol{a}_{\text{prev}}\|^2$ | $1.0 \cdot 10^{-2}$ |
| Table contact force | $-\|\boldsymbol{F}_{\text{c, table}}\|^2$ | $1.0 \cdot 10^{-5}$ |
| Package velocity | $-\|{}_{\mathcal{I}}\dot{\boldsymbol{r}}_{\text{package}}\|^2$ | $1.0 \cdot 10^{-2}$ |

Table 2: Observations

| | |
|---|---|
| **Robot-related Observations** | |
| ${}_{\mathcal{B}}\dot{\boldsymbol{r}}_{\text{base}} \in \mathbb{R}^3$ | Linear base velocity |
| ${}_{\mathcal{B}}\boldsymbol{\omega}_{\text{base}} \in \mathbb{R}^3$ | Angular base velocity |
| ${}_{\mathcal{B}}\boldsymbol{g} \in \mathbb{R}^3$ | Projected gravity vector |
| $\boldsymbol{q}_{\text{legs}} \in \mathbb{R}^{12}$ | Joint configuration without wheels |
| $\boldsymbol{o}_{\text{hooks}} \in \mathbb{R}^4$ | Hook directions (for pull doors) |
| $\dot{\boldsymbol{q}} \in \mathbb{R}^{16}$ | Joint velocity |
| $\boldsymbol{a}_{\text{prev}} \in \mathbb{R}^{16}$ | Previous actions |
| **Door-related Observations** | |
| ${}_{\mathcal{C}}\boldsymbol{r}_{CH} \in \mathbb{R}^3$ | Relative door handle position |
| ${}_{\mathcal{C}}\boldsymbol{r}_{CH_{\text{init}}} \in \mathbb{R}^3$ | Relative initial door handle position |
| **Package-related Observations** | |
| ${}_{\mathcal{C}}\boldsymbol{r}_{CP} \in \mathbb{R}^3$ | Relative package position |
| ${}_{\mathcal{C}}\boldsymbol{r}_{CT} \in \mathbb{R}^3$ | Relative table position |
| ${}_{\mathcal{C}}\boldsymbol{r}_{CB} \in \mathbb{R}^3$ | Relative bin position |

Table 3: Randomization Parameters

| Uniformly Randomized Property | Mean $\mu$ | Range $\epsilon$ | Unit |
|---|---|---|---|
| Global friction coefficient | 0.75 | 0.75 | - |
| Robot position $(x, y)$ | 0 | 0.6 | m |
| Initial robot yaw angle | 0 | 1 | rad |
| Initial joint angle deviation | 0 | 1 | rad |
| Added robot mass | 0 | 10 | kg |
| Package mass | 1.375 | 1.0 | kg |
| Door torque offset $\boldsymbol{\tau}_{\mathrm{const}}$ | $\begin{bmatrix} 10 & 0 \end{bmatrix}^{\top}$ | $\begin{bmatrix} 10 & 0 \end{bmatrix}^{\top}$ | $\mathrm{N\,m}$ |
| Door spring coefficient $\boldsymbol{k}$ | $\begin{bmatrix} 0 & 5 \end{bmatrix}^{\top}$ | $\begin{bmatrix} 0 & 5 \end{bmatrix}^{\top}$ | $\dfrac{\mathrm{N\,m}}{\mathrm{rad}}$ |
| Door damping coefficient $\boldsymbol{d}$ | $\begin{bmatrix} 25 & 1 \end{bmatrix}^{\top}$ | $\begin{bmatrix} 25 & 1 \end{bmatrix}^{\top}$ | $\dfrac{\mathrm{N\,m\,s}}{\mathrm{rad}}$ |

actively direct the robot's gaze. A visual marker, further explained in Section A.2, specifies the position of the door handle. Consequently, the observation $_C\boldsymbol{r}_{CH}$ is only available if the marker is detected by a camera. Always passing the door handle observation in the simulation would therefore not capture the real system behavior. Instead, the observation is set to $\boldsymbol{0}$ if the visual marker leaves the camera's FOV. This way, the agent learns to approximately partition the observation space and reason about when it is necessary to observe the visual marker. The agent can develop behaviors to mitigate a lost observation and to actively keep the marker in the FOV. An illustration of the approach is provided in Fig. 7. Note that the second door-related observation $_C\boldsymbol{r}_{CH_{\mathrm{init}}}$ is not set to $\boldsymbol{0}$ because the initial door handle position is static with respect to the inertial frame. The observation can thus be bootstrapped with the onboard localization of the robot even if the visual marker leaves the FOV.

## A.2 Real-World Setup

We utilize AprilTags [45] to obtain task-related observations in the real world. The AprilTag system features a vision-based algorithm that determines the relative position and orientation of detected tags. Two visual markers attached to the door provide the relative door handle position observation $_C r_{CH}$. If the robot does not detect the tags, the observation is set to $\mathbf{0}$ to achieve the same behavior as in simulation. The initial door handle position observation $_C r_{CH_{\text{init}}}$ is determined by two markers attached to the door frame. We make use of the robot's onboard localization to obtain an observation even if the tags leave the FOV of the camera. AprilTags also provide relative positions of the package, bin, and table. We do not make use of the proposed FOV simulation for the package manipulation task for two reasons. Firstly, it increases the difficulty of learning the desired behavior because the robot tries to keep the package in the FOV by leaning over the bin and falling. Secondly, the package is kept in the FOV naturally until the package is dropped, rendering the additional FOV constraint unnecessary for this task.

Furthermore, we note a few limitations with the current experimental setup. To achieve a $100\,\%$ success rate in the series of real-world experiments, it was vital to get reliable door observations through the camera system. Especially for fast rotations, the used visual fiducial system suffered from image blur and low frame rates. Observations might also degrade over longer periods of time if the fiducials leave the FOV, as the robot then purely relies on its localization. With the remedies mentioned above, we were able to resolve these issues, but longer horizon tasks might need more careful considerations.

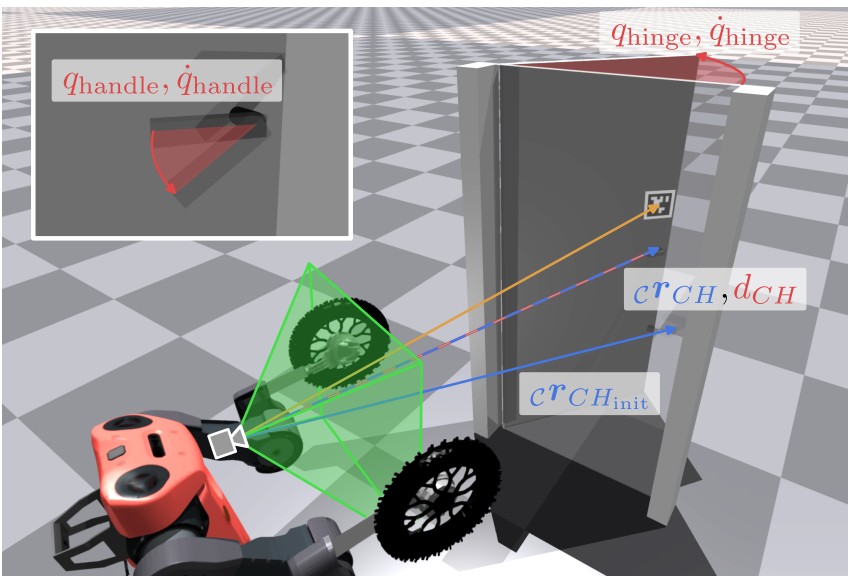

Figure 7: Door setup and FOV simulation. Components of the curiosity state $s_c$ are marked in red, while observations are marked in blue. The green cone represents the camera's FOV. A visual marker, attached to the door, is used to calculate the door handle observation $_C r_{CH}$. If the vector from the camera to the visual marker (orange) leaves the FOV cone, the door handle observation is set to $\mathbf{0}$.

### A.3 Quantitative Results

#### A.3.1 Comparison to a Naive Dense Reward Setting

To highlight the benefits of the curiosity-driven approach, we draw a comparison to a basic dense reward approach that involves comparable engineering effort.[3] For the door opening tasks, we define three dense rewards as guidance toward the sparse task reward. These rewards are defined as to minimize the distance from the wheel to the door handle, the door handle angle, and the door hinge angle. The hinge angle reward is clipped to ensure that the robot does not open the door too far since we consider the task only fulfilled if the door is opened within a specified angular window. We were able to tune the reward weights to deliver a similar performance to our approach for push doors, yielding an average success rate of 91 % in simulation, as seen in Fig. 8. We could not find weights that would result in successfully learning the pull door task. The package manipulation task is augmented with three dense rewards as well. These increase with de-

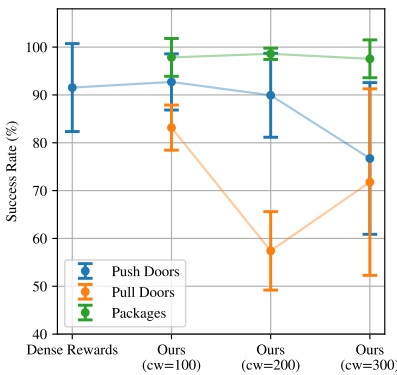

Figure 8: Success rate mean and standard deviation of successful training runs for different reward formulations, intrinsic reward weights (denoted as cw), and random seeds 1-10.

creasing distance between the right wheel and the right side of the package, the distance between the left wheel and the left side of the package, and the distance between the package and the bin. Again, we were unable to train a policy that achieves the desired task, as shown in Fig. 4.

#### A.3.2 Intrinsic Reward Scale Sensitivity

The training process shows a high level of robustness with respect to the scale of the intrinsic reward, as can be seen in Fig. 4. This can be explained by the reward's dynamic magnitude. At the beginning of training, the RND prediction error is large enough to overcome the local minimum imposed by shaping rewards. During training, the intrinsic reward shrinks and allows for optimization toward the task and shaping rewards. If the weight is chosen too large, the intrinsic reward might not decay enough such that the sparse reward, although discovered, might get overlooked in the optimization.

#### A.3.3 Robustness Against Environment Variation

We investigate success rates in multiple simulation experiments to analyze the robustness of learned policies against the variation of different environment parameters, including parameter values that are out-of-distribution of the learning tasks. We report the results in Fig. 9. The success rates are determined by observing 1000 differently randomized environments for one episode. First, the robot's initial position and orientation are randomized over a larger interval than during training. Second, the door handle height is randomized uniformly. Even though the handle height was not randomized during training, policies are able to adapt to different heights. Last, the package position and the bin position are randomized uniformly. Again, neither was randomized during training. The package position is randomized for a range up to $0.4\,\mathrm{m}$, as this covers the entire table surface. Trained policies exhibit a high level of robustness against the investigated disturbances.

---

[3]We also investigated curriculum learning as an alternative. The (naive) task curricula consisted of spawning the robot in favorable positions (e.g. very near to the door, with one wheel touching the door handle), and then gradually increasing the task difficulty by increasing the distance to the door. In our experiments, the robot was not able to successfully discover the desired behavior. We are convinced that the tasks could be solved with more intricate curricula, but these stand in no perspective with regard to the engineering effort to the proposed curiosity-driven approach.

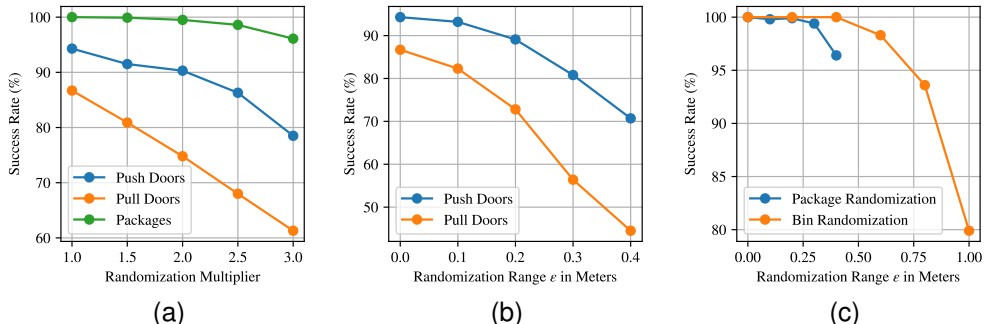

Figure 9: Simulation success rates for different out-of-distribution experiments. In (a), the initial robot position and yaw angle randomization range used during training is multiplied. In (b), the door handle height is randomized and in (c), the package and bin positions are randomized uniformly.

