# OpenReview forum: "Curiosity-Driven Learning of Joint Locomotion and Manipulation Tasks"
_robot-learning.org/CoRL/2023/Conference — CoRL 2023 Poster_

### Official Review · Reviewer_81RD · 2023-07-16

**Confidence:** 5
**Originality:** Fair
**Technical Quality:** Fair
**Clarity Of Presentation:** Good
**Impact:** 2

**Recommendation:**

Weak Accept: I recommend accepting the paper, but will not argue for my recommendation if the majority of other reviewers have a different opinion.

**Review:**

### Pros:
- Interesting tasks are studied in this paper including using a wheeled-legged robot to open a door.

### Cons:
- This paper does not include any quantitative evaluations. The only descriptions regarding the performance of the method are "over 16 times in a row without failure" for the door opening task and "over 6 times in a row without failure" for the package moving task. The quantitative metrics can include measurements of success rates, time to completion, and robustness to different environments and objects.
- It does not compare with any other baselines methods, including simple reward shaping, simple curriculum learning, or simple imitation.
- One of the key motivations of the paper is using exploration bonus to alleviate heavy reward engineering. However, it is shown in the appendix that for each task, excluding the RND bonus term, 9 reward terms are used. As a result, a new natural question to ask is that why not just add several more reward shaping terms (e.g. L2 distance to the handle, etc) to solve these locomotion-manipulation tasks?
- All the experiments focus on the exactly same sim and real pairs. For the door-opening task, the robot resets to the exact same initial position very close to the door, and the door is the same in all experiments. The same applies to the package-moving task. Hence, it makes me wonder does the method just overfit to a single environment without generalizing to other environments?
- No limitations regarding the tasks, real-world robotics challenges or sim-to-real transfer are described in the limitation section. The limitation section solely focuses on describing the difficulties of using the RND exploration bonus, which is well-known.
- This paper neglects most of (if not all) the prior work on manipulation-locomotion and locomotion-manipulation from the robotics community, The prior work section overwhelmingly focuses on exploration and RL training. However, this work does not introduce any new or change to the existing exploration method RND. In contrast, many existing works on manipulation-locomotion and locomotion-manipulation are not mentioned at all. These include but are not limited to:
  - Human Motion Control of Quadrupedal Robots using Deep Reinforcement Learning. Sunwoo Kim, Maks Sorokin, Jehee Lee, Sehoon Ha. RSS 2022.
  - Deep whole-body control: learning a unified policy for manipulation and locomotion. Zipeng Fu, Xuxin Cheng, Deepak Pathak. CoRL 2022.
  - Hierarchical Reinforcement Learning for Precise Soccer Shooting Skills using Quadrupedal Robots. Yandong Ji, Zhongyu Li, Yinan Sun, Xue Bin Peng, Sergey Levine, Glen Berseth, Koushil Sreenath. IROS 2022.
  - Legs as Manipulator: Pushing Quadrupedal Agility Beyond Locomotion. Xuxin Cheng, Ashish Kumar, Deepak Pathak. ICRA 2023.
  - Cascaded Compositional Residual Learning for Complex Interactive Behaviors. K. Niranjan Kumar, Irfan Essa, Sehoon Ha.
  - DribbleBot: Dynamic Legged Manipulation in the Wild. Yandong Ji, Gabriel B. Margolis, Pulkit Agrawal. ICRA 2023.
  - Creating a Dynamic Quadrupedal Robotic Goalkeeper with Reinforcement Learning. Xiaoyu Huang, Zhongyu Li, Yanzhen Xiang, Yiming Ni, Yufeng Chi, Yunhao Li, Lizhi Yang, Xue Bin Peng, Koushil Sreenath.
  - Learning agile soccer skills for a bipedal robot with deep reinforcement learning. Tuomas Haarnoja, et al.
  - Multi-Agent Manipulation via Locomotion using Hierarchical Sim2Real. Ofir Nachum, Michael Ahn, Hugo Ponte, Shixiang Gu, and Vikash Kumar.
  - Contact optimization for non-prehensile loco-manipulation via hierarchical model predictive control. Alberto Rigo, Yiyu Chen, Satyandra K Gupta, Quan Nguyen. ICRA 2023.

**Quality Of The Limitations Section:**

Limitations are not well addressed

**Questions For Rebuttal:**

- Can quantitive results be included? For both simulation and real world.
- Can some environmental diversities be shown to validate that this method does not just overfit to a single task, environment, and initial state?
- Can authors include a much more comprehensive discussion on the similarities and differences with the prior work that I listed in the above section?
- Can comparisons with baselines (e.g. reward shaping, curriculum learning, etc) be included?
- Can the limitation section include more related information about when the method fails? For example, difficulties in relying on a fiducial system, shown in the the supplementary material: "We make use of the robot’s onboard localization to obtain an observation even if the tags leave the FOV of the camera".

**Robotics Focus:**

Sufficient demonstration on hardware

**Summary Of Paper:**

This paper proposes to use curiosity-driven reward bonus to learn loco-manipulation tasks. A wheeled-legged robot is used in biped mode for experiments in the simulation and the real world on two tasks: door opening and package manipulation. Concretely, the vanilla RND is used as the exploration bonus. The authors also analyze the emerging behaviors and the sensitivity of the learning process.

**Summary Of Recommendation:**

My recommendation reflects the current problems of the submission. Would be very happy to see future changes, and I'll update my score accordingly.

---

### Official Review · Reviewer_a9Bt · 2023-07-19

**Confidence:** 4
**Originality:** Good
**Technical Quality:** Very Good
**Clarity Of Presentation:** Very Good
**Impact:** 4

**Recommendation:**

Weak Accept: I recommend accepting the paper, but will not argue for my recommendation if the majority of other reviewers have a different opinion.

**Review:**

The curiosity driven approach, RND, is successfully applied and validated on the two real robot tasks in this paper. Combining with the curiosity state, the proposed method show task-agnostic properties and generalize well to two different tasks.

Strength: Results on the real robot are pretty impressive. The motivation of the paper is clear. The task-agnostic properties of the proposed method is thoroughly discussed and convincing. Results analysis of the experiments results are comprehensive and insightful. The paper also includes lots of implementation details regarding weights design and normalization with detailed discussion, which are very useful for the community. The emergence of different behaviors and its pros and cons are very interesting.

Weakness:
(1) The trained policy looks aggressive on real robot demonstration even with the shaping reward. It seems the shaping rewards may still require extensive tuning as it still contains task specific terms.
(2) As mentioned by the author, the shaping rewards bring difficulty to the skill discovery. A reward scaling scheme is applied to compensate this. However, it’s unclear if this is sufficient for more complicated systems or longer horizon tasks.
(3) There is no comparison between the proposed method and other baselines. On line 211, “The need for guided exploration can be seen in a training run with extrinsic reward signals only”. I couldn’t find this run in the paper or in the supplement.


**Quality Of The Limitations Section:**

Limitations are addressed clearly

**Questions For Rebuttal:**

Since the shaping rewards bring difficulty to the skill discovery. Have the author considered fine-tuning the policy with the shaping reward later by combining BC and RL? For example, policy 1 can be learned without the shaping reward. Policy 2 can be learned by using policy 1 as the teacher policy with BC loss and using the shaping reward as the RL loss.

On line 211, “The need for guided exploration can be seen in a training run with extrinsic reward signals only”. I couldn’t find this run in the paper or in the supplement.


**Robotics Focus:**

Sufficient demonstration on hardware

**Summary Of Paper:**

This paper studies the problem of learning joint locomotion and manipulation tasks. Using task-specific rewards can require labor intensive engineering and require re-tuning for a new task. An intrinsically motivated RL approach using Random Network Distillation (RND) is proposed to reduce task-specific engineering with sparse task rewards and intrinsic rewards. The method is shown to generalize to the door opening task and the packing task. Learned policies are evaluated on a wheeled-legged robot on two tasks and achieve 100% success rate.


**Summary Of Recommendation:**

After reading the updated paper and the rebuttal, most of the question I asked above is solved. My recommendation remains the same. The paper proposed an approach applying RND on learning joint locomotion and manipulation tasks. Comprehensive results analysis and discussion show the task-agnostic properties of the method. Some implementation details and insights are valuable to the community. Demonstration of the learned policies on the hardware is impressive.

---

### Official Review · Reviewer_qvEX · 2023-07-20

**Confidence:** 4
**Originality:** Poor
**Technical Quality:** Fair
**Clarity Of Presentation:** Very Good
**Impact:** 3

**Recommendation:**

Weak Accept: I recommend accepting the paper, but will not argue for my recommendation if the majority of other reviewers have a different opinion.

**Review:**

These days, training RL policies for robot control often involve complex, dense, heavily-shaped and task-specific rewards in order to solve the task. While the resulting behaviours are generally impressive and robust, this reward design requires significant tuning and effort for every task under consideration. Approaches that allow "coarse" reward definition by improving exploration behaviour are hence incredibly relevant for the robotics community. While I commend the authors for the clear presentation, professionally-montaged supplementary video and the hardware results, I have a couple of major concerns with this work:
- There has been lots of prior work (e.g. https://arxiv.org/abs/1708.02190 from 2017) on applying intrinsic motivation to robotics settings. It would be appreciated to focus the related work section on this intersection and explicitly contrast the proposed approach.
- The methodology is very incremental from base RND, and dare I say even removes some of the appeal by requiring only task-relevant state features to be used for the intrinsic reward. This effectively shifts some of the effort from reward design to feature design. This may be easier or less sensitive, but from the results in the paper it is hard to judge.
- In a similar vain, a major issues with intrinsic exploration methods such as the one proposed, is the balance between intrinsic and task rewards, and in this case also regularisation rewards. This was also discussed in related work such as https://arxiv.org/abs/1903.08542. The authors touch on this and even note that it is important to tune reward scales, even with which the learned behaviours are still very dynamic (underscaling the regularisation reward). I would like to see more results on the sensitivity of reward scales to being able to solve the task and on the resulting behaviour.
- While the behaviours shown in the figures and video are reasonably novel and interesting, I find the paper lacking quantitative results, only present in the caption of Figure 4 (4/6 seed success, missing from main text) and Section 5.2. There are no baselines that are compared to (e.g. other exploration mechanisms, dense rewards, model-based?). The evaluations should at least mention an overall task success rate in sim across many trials, and ideally so should the results in real (when the agent succeeds in 6 trials in a row in real on the package manipulation task, does that mean it failed on the 7th? Were only 6 trials executed?). While its great to see that the resulting controllers can be deployed on hardware, this is using existing approaches so in itself does not validate the proposed approach.

**Quality Of The Limitations Section:**

Limitations are addressed clearly

**Questions For Rebuttal:**

- What is the overall success rate for both tasks and across seeds in simulation, and ideally in real?
- What is meant with "succeeds 6 times in a row"? Did it fail on the 7th? Were only 6 trials performed?
- How sensitive are the results, quantitatively, to the scales of the various rewards?

**Robotics Focus:**

Sufficient demonstration on hardware

**Summary Of Paper:**

The authors propose a method for learning loco-manipulation skills with sparse rewards. They employ intrinsic motivation in the form of Random Network Distillation to provide a guiding signal for the agent to discover the sparse reward effectively. The based RND method is modified in two ways. First, a subset of task-relevant state features is used instead of the full observation. Second, the inputs for the RND networks are continuously normalised. The approach is successfully applied to both a door opening and package manipulation task with a wheeled-legged robot, and the resulting controllers are successfully transferred to hardware.

**Summary Of Recommendation:**

As the paper lacks quantitative results and the proposed method is very incremental compared to prior work, with no significant discussion of such, I find it difficult to vote to accept this paper. I'm voting for a weak reject instead of a strong reject due to the excellent presentation and the results on hardware.

Edit after rebuttal: I have raised my recommendation after the authors' rebuttal due to the additional quantitative results and the rephrasing of the contributions. I would still urge the authors to move the quantitative results to the main text instead of appendix.

---

### Official Review · Reviewer_bDmA · 2023-07-23

**Confidence:** 4
**Originality:** Fair
**Technical Quality:** Good
**Clarity Of Presentation:** Good
**Impact:** 3

**Recommendation:**

Weak Accept: I recommend accepting the paper, but will not argue for my recommendation if the majority of other reviewers have a different opinion.

**Review:**

The paper has following strengths:

- The tasks as well as the real robot setup considered in this work are interesting, novel, and likely significant for future works.
- The real-world robot demos are impressive.
- The paper writing is clear and effectively conveys the contributions as well as the results of the work.

Despite the strengths, there are a few major weaknesses:

- While the tasks considered are interesting, the technical contributions of this work might be limiting. The main technical contribution of the paper seems to be an adaptation of the previously proposed RND method, where instead of using full observation, the paper proposes a new notion of “curiosity state”, which is a subset of full environment state and needs to be hand-selected for each task. By manually selecting the curiosity state to be a few “important parameters”, this essentially forms an implicit task curriculum that encourages the agent to explore within the manifold of those parameters. This does not seem to be an improvement of RND, where the whole premise of intrinsic reward is to reduce task-specific engineering.
- To demonstrate technical contributions, an important and necessary way is to compare to prior works quantitatively, but this also is lacking from the paper. Except for the training plot that shows the progression of task-specific reward and intrinsic reward, no other quantitative results are seen from the paper.
- Given the impressiveness of the real-world setup, the paper can also be shown to be of good value to the community if extensive experimental results are shown (e.g., across many more tasks and benchmarked against baselines), but the experimental results currently  in this work are lacking from this perspective.

**Quality Of The Limitations Section:**

Limitations are addressed clearly

**Questions For Rebuttal:**

- Section 5.1 mentions that training with extrinsic reward would fail, but no information or experiment of this can be found in the paper. Can this be further clarified?

**Robotics Focus:**

Sufficient demonstration on hardware

**Summary Of Paper:**

The paper tackles the challenging joint locomotion and manipulation tasks — door opening and package manipulation — on a wheeled-legged robot. The paper proposes to train RL policies with an intrinsic reward to encourage exploration, where the intrinsic reward is adapted from previously proposed Random Network Distillation (RND). The simulation-trained policy is shown to transfer to real-world setup.

**Summary Of Recommendation:**

The paper tackles important and interesting tasks of joint locomotion and manipulation on wheeled-legged robot, while showing impressive real robot demos resulted from an RL policy trained purely in simulation. However, the paper lacks quantitative evaluations which make the technical contributions or experimental contributions unclear.

---

### Decision · Program_Chairs · 2023-08-30

**Decision:**

Accept (Poster)

**Comment:**

There are strengths and weaknesses of the paper. For the strengths, the qualitative real robot results are quite nice. The biggest weakness before the rebuttal was the lack of quantitative results, though the quantitative results are much more thorough after the rebuttal. Now, it seems like the main limitations are the limited "generalization to different environments/scenes (testings are done on one door, one object, and relying on ARTags)" (reviewer 81RD), the significant revision needed to incorporate the new results (reviewer bDmA), and the limited technical contribution (multiple reviewers).

Overall, I think the strengths outweigh the weaknesses and recommend acceptance.